# LDPE Transformation by Exposure to Sequential Low-Pressure Plasma and TiO_2_/UV Photocatalysis

**DOI:** 10.3390/molecules26092513

**Published:** 2021-04-26

**Authors:** Luis D. Gómez-Méndez, Luis C. Jiménez-Borrego, Alejandro Pérez-Flórez, Raúl A. Poutou-Piñales, Aura M. Pedroza-Rodríguez, Juan C. Salcedo-Reyes, Andrés Vargas, Johan M. Bogoya

**Affiliations:** 1Laboratorio de Microbiología Ambiental y Suelos, Grupo de Biotecnología Ambiental e Industrial (GBAI), Departamento de Microbiología, Facultad de Ciencias, Pontificia Universidad Javeriana, Bogotá 110-23, DC, Colombia; apedroza@javeriana.edu.co; 2Grupo de Películas Delgadas y Nanofotónica, Departamento de Física, Facultad de Ciencias, Pontificia Universidad Javeriana, Bogotá 110-23, DC, Colombia; cjimenez@javeriana.edu.co (L.C.J.-B.); salcedo.juan@javeriana.edu.co (J.C.S.-R.); 3Grupo de Investigación en Fitoquímica (GIFUJ), Departamento de Química, Facultad de Ciencias, Pontificia Universidad Javeriana, Bogotá 110-23, DC, Colombia; alejandroperez@javeriana.edu.co; 4Laboratorio de Biotecnología Molecular, Grupo de Biotecnología Ambiental e Industrial (GBAI), Departamento de Microbiología, Facultad de Ciencias, Pontificia Universidad Javeriana, Bogotá 110-23, DC, Colombia; rpoutou@javeriana.edu.co; 5Grupo de Física Matemática, Departamento de Matemáticas, Facultad de Ciencias, Pontificia Universidad Javeriana, Bogotá 110-23, DC, Colombia; a.vargasd@javeriana.edu.co; 6Independent Researcher, Bogotá, DC, Colombia; johanbogoya@gmail.com

**Keywords:** low-density polyethylene (LDPE), direct-current low-pressure plasma, TiO_2_/UV photocatalysis

## Abstract

Low-density polyethylene (LDPE) sheets (3.0 ± 0.1 cm) received sequential treatment, first by the action of direct-current low-pressure plasma (DC-LPP) with a 100% oxygen partial pressure, 3.0 × 10^−2^ mbar pressure, 600 V DC tension, 5.6 cm distance, 6-min treatment. Then, sheets were submitted to TiO_2_ photocatalysis at UV radiation at 254 nm (TiO_2_/UV) with a pH value of 4.5 ± 0.2 and a TiO_2_ concentration of 1 gL^−1^. We achieved a complementary effect on the transformation of LDPE films. With the first treatment, ablation was generated, which increased hydrophilicity. With the second treatment, the cavities appeared. The changes in the LDPE sheets’ hydrophobicity were measured using the static contact angle (SCA) technique. The photocatalytic degradation curve at 400 h revealed that the DC-LPP photocatalysis sequential process decreased SCA by 82°. This was achieved by the incorporation of polar groups, which increased hydrophilicity, roughness, and rigidity by 12 and 38%, respectively. These sequential processes could be employed for LDPE and other material biodegradation pretreatment.

## 1. Introduction

On an everyday basis, one of the most employed polymers worldwide is low-density polyethylene (LDPE) with industrial, commercial, and domestic uses. Low-density polyethylene transformation and/or biotransformation is difficult and slow because it is a high molecular weight polymer with hydrophobic carbonated chains and low nitrogen content, thus generating a major environmental problem affecting the use of water, resources, soil, and air. During LDPE incineration and its subsequent disposal in sanitary landfills, gaseous emissions, such as COx and volatile organic compounds (VOCs) are generated [1,2]. Therefore, to promote LDPE degradation different physical, chemical, and biological treatments can be implemented, such as direct-current low-pressure plasma (DC-LPP), photocatalytic oxidation, photo-oxidation, thermo-oxidation, and biological transformation [3,4]. Most of these treatments are performed in the amorphous region of the polymer to modify LDPE’s hydrophobic surface into a hydrophilic one and generate low molecular weight carbonated fractions (chain hydrolysis and formation of vinyl and carbonyl groups) [1,5]. In such a way its subsequent transformation is facilitated by implementing other technologies or its final disposal or recycling is carried out.

DC-LLP modifies LDPE surface (ablation) by randomly inserting onto the plastic’s surface carboxyl polar groups (-COOH) and hydroxyl groups (-OH) [6,7] that generate crosslinking with the polymer’s carbon chains [8,9], forming oxidized low molecular weight structures that become more susceptible for subsequent transformation [10,11,12]. To generate plasma through electric discharge, inert gases are employed, such as argon (Ar) [13,14] and reactive gases such as oxygen (O_2_) [3,7,15] or a mix of both [8,11,16]. In the presence of O_2_, plasma action results in double bond formation in LDPE’s polymeric chain, with groups such as vinyl and vinyldenes (R_1_R_2_C=CH_2_) with hydrogen release (H_2_) or with the participation of hydrogen in a second reaction to form water (H_2_O) [16]. Vinyl group formation is also possible without H_2_ release, where its formation occurs with the participation of methyl groups (CH_3_-) present in LDPE [11]. At the same time, organic hydroperoxides are formed (CH_3_-O-OH) on the material’s surface, due to the ease of breaking O-O bonds that later form free radicals, such as carboxy (C-O•), carboxyl (COO•) [7], hydroxyl (OH•), methyl (CH_3_•), or alkoxy (CH_3_-O•) [17] on the surface of the treated material [18]. Last, due to the elimination of H_2_ atoms from the polymeric chain, various polar groups are formed on the LDPE surface, such as (hydroxyl (OH-), carbonyl (C=O), and carboxyl (C=OOH)) [19,20,21]. These groups contribute to LDPE’s increased surface hydrophilicity [17].

On the other hand, advanced oxidation processes, such as photocatalysis with TiO_2_, are a physical/chemical alternative to modify or transform different types of plastic [22,23,24,25]. The photocatalytic process with TiO_2_ begins when the oxide semiconductor is electromagnetically irradiated (λ < 400 nm) and a photoelectron excitation takes place, where photon energy (*hv*) is absorbed by the electron in the TiO_2_ valence band (VB), which causes energy transitions between the VB and the conduction band (CB) to ensue. This dynamic generates vacancies or holes in the spaces previously occupied by VB electrons, producing an electron–hole pair (e^−^/h^+^) [6]. Electron–hole pairs (e^−^/h^+^) can produce a series of oxidation–reduction reactions, where photogenerated e^-^ in CB can form O_2_•^−^, and the photogenerated h^+^ in VB can form OH•. Likewise, the photogenerated holes initiate oxidative stages where adsorbed organic molecules on the material’s surface can be oxidized to become CO_2_ and water. Relating to plastic photocatalytic oxidation, various authors propose reactive oxygen species (ROS) randomly oxidize LDPE carbon chains, in the amorphous phase as well as the crystal phase [22,25,26].

Both DC-LLP and photocatalysis modify the LDPE’s chemical structure and mechanical properties, favoring its degradation, yet a combined action between both will notably increase the material transformation. Currently, this combination has not been studied. Therefore, the objective of this study was to select reaction conditions and/or operating conditions to perform plasma discharge and TiO_2_/UV photocatalysis as technologies to transform LDPE sheets.

## 2. Results and Discussion

### 2.1. Initial LDPE Sheet Characterization

Clean 3 cm^2^ LDPE sheets (pristine LDPE) results before being submitted to experimentation appear in Table 1.

### 2.2. Plasma Discharge on LDPE Sheets–Static Contact Angle

In previous work by Gómez-Méndez et al. [29], O_2_ plasma was used for LDPE pretreatment for consecutive biodeterioration with *P. ostreatus*. In this work, it was demonstrated that plasma discharge favored the material’s hydrophobicity and subsequent colonization and biodeterioration by the fungus [29]. In the present work, argon (Ar) in addition to O_2_, was used for plasma discharge since various authors have described its use for LDPE surface modification [30,31,32].

Regarding preliminary assays, it was observed that only 24.8% (32 treatment combination) met with conditions 1 and 2 previously described in the methodology (Appendix A). Out of the 32 combinations, 15% corresponded to 100% O_2_ discharge, 41% to 100% Ar plasma discharge, and 44% to the (1:1) mix of gases discharge (Appendix A).

For discharge of 100% Ar (*p* = 0.0126) (Appendix A), the two best treatments were 2.4 × 10^−2^ mbar, 900 V treatment and 2.4 × 10^−2^ mbar, 800 V, presenting a static contact angle (SCA) of (26 ± 5)° and (30 ± 5)°, respectively (Figure 2A). For 100% O_2_ (*p* = 0.4275) (Appendix A), the greatest decrease in SCA was achieved with 3.0 × 10^−2^ mbar, 700 V with (20 ± 4)° treatment followed by the 3.0 × 10^−2^ treatment mbar, 600 V with (21 ± 3)° (Figure 1B). The mixed discharge treatment Ar-O_2_ (1:1) (*p* = 0.0001), (Appendix A) showed the greatest decrease in SCA for 3.0 × 10^−2^ mbar, 600 V (SCA 28 ± 2)° followed by 2.6 × 10^−2^ mbar, 700 V with an SCA of (30 ± 2)° (Figure 1C). The comparison of means between treatments (*p* = 0.0139) (Appendix A) showed that the treatments with the highest hydrophilicity (lower SCA) were 100% O_2_, 3.0 × 10^−2^ mbar, 700 V (20 ± 4)° followed by 100% O_2_ treatment, 3.0 × 10^−2^ mbar, 600 V (21 ± 3)°. However, no significant differences were observed between them (*p* = 0.9981). Therefore, the lowest voltage treatment was used (Figure 1D).

The main difference between Ar And O_2_ discharge is that Ar results in chain cleavage and bond rupture on LDPE’s surface, generating Alkyl radicals (C•) by splitting of C-H and C-C bonds. These radicals prefer to interact with adjacent polymer chains rather than with polar groups, forming crosslinks [33,34], while O_2_ discharge is very reactive and efficient in incorporating the surface polar groups, such as carbonyl (C=O), carboxyl (C=O-OH), and hydroxyls (-OH), which are responsible for decreasing SCA due to hydrophobicity changes [35,36]. Thus, in the present work, the highest hydrophilicity values are those for O_2_ discharge. In addition, once the discharge ends, C• on the surface can react with atmospheric O_2_ [30], and when removed from the reaction chamber, they oxidize the material and increase the presence of (C=O), (C=O-OH), esters (C=O-O-R) and (-OH) [14,37]

Abou Rich et al. [30] describe LDPE exposed to the environment after plasma discharge generates O• and hydroxyl radicals (OH•) that tend to take secondary hydrogens from the LDPE chains, resulting in the formation of alkyl radicals (CH_3_•) [30]. These radicals can react with atomic oxygen (O_2_) or ozone (O_3_), allowing the formation of alkoxy (C-O•). Furthermore, LDPE-carbonated radicals can react with O_2_ radicals from the environment and form peroxide (C-O-O•) and hydroperoxide (H-O-O•) radicals [34].

Guruvenket et al. [38] observed when polyethylene (PE) was submitted to Ar and O_2_ plasma discharge at different voltages and time of discharge, SCA decrease was greater in comparison with only O_2_ plasma treatment. Likewise, they described how the time/voltage ratio required to decrease SCA was lower for O_2_ in comparison with Ar discharge [38]. Ar–O_2_ plasma forms oxidized structures crosslinked with LDPE’s surface, suggesting a more rapid transformation due to the incorporation of high-density functional groups [39]. In the present work, the highest SCA decrease was observed for treatments with O_2_ discharge with lower voltages (600 and 700 V) in comparison with Guruvenket et al. [38], as was described by Gómez-Méndez et al. [29].

Generally, LDPE suffers non-permanent physical and chemical modifications on its surface during and after plasma discharge [11]. Plasma-irradiated surface tends to reverse changes during discharge and return to its original chemical state [40,41] in a phenomenon known as “hydrophobic recovery” [12,33]. Hydrophobic recovery after the best O_2_ plasma discharge treatment is illustrated in Figure 2B; at day 0, SCA was at zero degrees and seven days later it was at (65 ± 9)°, representing a 75.4% hydrophobic recovery (stored at 14 °C), while control without plasma discharge remained at (86 ± 3)° (Figure 2A). Pandiyaraj et al. [33] refer to LDPE hydrophobic recoveries of 82% after oxygen discharge treatment and 15 days of storage at RT. Sanchis et al. [17] report 30° recovery after 170 h of storage with 30-min O_2_ plasma treatment. On the other hand, Abou Rich et al. [30] described SCA as close to 80°, even after 60-days post-treatment with oxygen plasma.

Hydrophobic recovery can occur through different mechanisms: (a) the reorganization of the polymer’s surface when low and high molecular weight molecules, not modified by the discharge, diffuse from the material’s core to the outer layers [34]; (b) diffusion of low molecular weight oxidized molecules, and (c) reorientation of chemical polar groups towards the core of the polymer [8,12,42]. According to Morent et al. [43], the degree of crosslinking affects hydrophobic recovery (ageing). The greater the crosslinking of LDPE, the more difficult chain movement will be, reducing the effect of post-plasma ageing [43].

The static contact angle for LDPE treated with O_2_ plasma for 6 min and control are in Figure 2A (ablation curve). An SCA decrease occurred for sheets submitted to discharge as a function of time until reaching zero (0°) after 4 min, while the SCA for the control was θ = (89 ± 2)°; demonstrating O_2_ plasma discharge under the conditions was effective in increasing the material’s hydrophilicity. Hydrophobic recovery after treatment appears in Figure 2B. One day after discharge, SCA was at zero degrees (0°) and day 7 after storing at 14 °C, SCA was at (65 ± 9)°, representing a 75.4% recovery. The control without discharge remained at (86 ± 3)°.

### 2.3. Other Response Variable Associated with Plasma Discharge

The following results were obtained from LDPE sheets submitted to the best treatment: (100% O_2_ 3.0 × 10^−2^ mbar and 600 V) SCA (21 ± 3)°, final weight (4.2 ± 0.1) mg, roughness (10 ± 3) nm, Young´s modulus (41 ± 6) MPa, and yield strength (11 ± 2) MPa. In comparison, pristine LDPE-treated material presented a 76% SCA decrease (*p* = 0.0139) with a 100% increase in roughness, whereas Young´s modulus and yield strength increased by 21% and 12%, respectively, but not significantly in comparison with pristine LDPE sheets (*p* = 0.05234).

Complete Fourier-transform infrared spectroscopy (FTIR) spectra for pristine LDPE sheets (black line) and post-plasma treatment with 100% argon (red line), argon–oxygen gas mix (brown line), and 100% oxygen (blue line) appear in Figure 3. Complete spectra characteristic of LDPE sheets presented signals at 2920 cm^−1^ (CH_2_ asymmetric stretching), 1460 cm^−1^ (CH_2_ band bending), and 729 cm^−1^ (CH_2_ rocking deformation). It is important to highlight that a decrease in these signals appeared after treatments (Figure 3B–D). For the 100% O_2_ treatment, the decrease in the signal at 906 cm^−1^ was observed, corresponding to vinyl groups, and signals at 975 cm^−1^ and 1180 cm^−1^ appeared, indicating peroxide groups and alcohol bonds, respectively (Figure 3E). Figure 3B illustrates a decrease in the signal at 2920 cm^−1^, which can be attributed to differences in the material’s width, caused by the ablation process [27]. Rajandas et al. [28] describe how the 2920 cm^−1^ signal is directly proportional to LDPE’s concentration, which could indicate that the material exposed to O_2_ plasma discharge when presenting a higher intensity would involve a loss of mass [28]. The same behavior was observed in Figure 3C,D. Shi et al. [44] specified that plasma active species, such as ions and electrons, bombard LDPE’s surface macromolecular chains, generating a cleavage in the C-H and C-C bonds producing oligomers and small organic molecules; hence, increasing loss of mass [44]. Shikova et al. [3] described O_2_ plasma discharge on PE unsaturated vinyl groups (-CH=CH2), vinylenes (R1R2C=CH2), and trans vinylenes (-CH2=CH2-), whose absorption bands appear between 890 and 960 cm^−1^ (Figure 3E). In the present study, in the vicinity of 906 cm^−1^, presence of vinyl was observed in the pristine LDPE, as well as Ar and Ar–O_2_ mixed treatment, which could suggest that LDPE sheets employed in this study contained impurities of unsaturated nature. After, 100% O_2_ plasma treatment bands in the 1180 cm^−1^ and 975 cm^−1^ range appeared, associated with alcohol and peroxide groups, respectively [45]. These results suggest LDPE’s sheet double bond loss and gain of carbonyl and peroxide groups after O_2_ plasma treatment. Exposure of LDPE sheets to plasma produces mainly radicals, which are converted into peroxide radicals, subsequently transformed by auto-oxidation and/or crosslinking into hydroperoxides [18]. Peroxide radical formation takes place mainly in the amorphous region due to oxygen high diffusion capacity [8].

LDPE’s morphological surface changes after plasma treatment were evidenced by SEM, whereas atomic force microscopy (AFM) revealed roughness results. Pristine LDPE surface without treatment is illustrated in Figure 4A. The pristine LDPE (Figure 4A) presented a homogeneous surface, while the sheets exposed to O_2_ (100%) exhibited much more change (Figure 4D) due to the presence of cracks and fissures on the surface (ablation). The AFM data revealed that the average roughness of the pristine LDPE was (5 ± 1) nm, while that of the 100% O_2_ treatment was (10 ± 3) nm (*p* = 0.0196). As determined by AFM after discharge, LDPE’s surface roughness changes doubled in comparison with pristine LDPE. Thus, this demonstrates the existence of processes leading to the material’s ablation, which removed the surface layers of the material [10]. These changes depend on the time of plasma exposure and voltage [32]. In this work, LDPE’s roughness increased by (5 ± 4) nm under 0.03 mbars of pressure with plasma generated at a cathode distance of 5.6 cm and six minutes of O_2_ gas exposure.

As shown by scanning electron microscopy (SEM) images, the greatest change was observed for O_2_ treatment (Figure 4D), followed by Ar–O_2_ mix (Figure 4C), and Ar last (Figure 4B). Pandiyaraj et al. [33] achieved greater roughness with an Ar–O_2_ plasma mix treatment for 10 min, 0.2 mbar pressure, and a 5 cm cathode distance. Švorčík et al. [10] reported an increase in PE roughness from 4 nm to 7 nm s after Ar plasma. Ataeefard et al. [11] reported roughness changes after exposing LDPE to Ar, O_2_, N_2_, and CO_2_ plasma discharge. Moreover, Sanchis et al. [17] observed an increase in roughness from 21 nm to 30 mm after submitting LDPE sheets to 30 min of O_2_ discharge. According to Ataeefard et al. [42], defined surface morphology can vary depending on the discharge voltage, time of exposure, and the type of gas employed [42]. According to Sanchis et al. [17], roughness augments LDPE’s wettability. In the present study, O_2_ plasma discharge increased LDPE roughness five-fold and generated greater hydrophilicity (four-fold higher than pristine LDPE), demonstrating O_2_ plasma discharge was more effective in generating ablation processes on LDPE’s surface in comparison with Ar discharge.

For the viscoelastic properties, Young’s modulus and yield strength values increased after O_2_ plasma discharge. An augmentation in LDPE’s viscoelastic properties suggests it becomes more rigid after treatment due to two facts; crosslinking between chains generated by plasma discharge [9] and polar group incorporation [11], restricting their relative movement, enhancing the polymer’s resistance. Other reasons for the increase in material stiffness are that LPDE is a semi-crystalline polymer, and after being plasma exposed (under the conditions of this study), the crystalline zone remained unchanged, but the weak bonds of the amorphous region were lost.

### 2.4. TiO_2_/UV Photocatalysis

#### 2.4.1. Commercial TiO_2_ Characteristics

Figure 5 depicts SEM images at different magnifications (5000× and 10,000×). At 5000×, a uniform material was observed with 1.23 ± 0.19 µm particle size. Observed spaces corresponded to cracks or sites where no TiO_2_ was deposited (Figure 5A). At greater magnification (10,000×), formation of aggregates composed of different particle sizes (polydisperse material) and forms (polymorphic) was observed, while spherical shapes predominated in Aldrich brand TiO_2_ (Figure 5B).

EDS analysis revealed that the Aldrich™ TiO_2_ contained titanium (Ti) (8.90 wt%) and oxygen (O_2_) (89.62 wt%) (Figure 5D). On the other hand, the XRD profile (Figure 5C), shows how the peak with the highest intensity is the orientation (A101), which corresponds to the anatase phase of TiO_2_ [46,47].

#### 2.4.2. 2^2^ Factorial Design and Photocatalysis at 300 h

According to Appendix A, the static contact angle, final weight, Young’s modulus, and yield strength were significant (*p* < 0.0001). Therefore, the main factors and interactions among factors were evaluated. For SCA, the R^2^ was 0.8223, while for the final weight it was 0.8795. For the viscoelastic properties of Young’s modulus and the yield strength, the R^2^ were 0.9800 and 0.8100, respectively. There was a high correlation among predicted and observed values with precision values greater than 4.0, confirming results were not the result of experimental noise (Appendix A).

pH and TiO_2_ concentration were factors that influenced SCA (*p* < 0.0001), with contributions of 45% and 40%, respectively. Additionally, standardized effects with their respective signs were +43.6, −6.5, −6.1, and +0.5; indicating the aforementioned factors could be employed at their low level. pH and TiO_2_ concentration were also influential factors (*p* < 0.0001 and *p* = 0.0002, respectively); regarding LDPE sheet weight loss, the highest contribution percentage was obtained with pH (60%) (Appendix A).

On the other hand, LDPE sheet viscoelastic properties were included in the statistical analysis (Young’s modulus and yield strength), which were significant (*p* < 0.0001). For Young’s modulus, pH and TiO_2_ concentration and their interaction were significant (*p* < 0.0001, obtaining 33% contribution percentages (Appendix A). These results indicate a decrease in Young’s modulus could be favored by low levels of these factors. Last, for yield strength, the most influential factor was pH (factor A, *p* < 0.0001), followed by AB interaction (p = 0.0007), and TiO_2_ concentration (factor B, *p* = 0.026), with contribution percentages of 70, 20, and 30%, for factor A, B, and AB interaction, respectively. Again, it was observed that a decrease in yield strength was favored by low levels of each factor and even their interaction was employed (Appendix A).

Mean comparison among treatments demonstrated significant differences for SCA (*p* = 0.0032), final weight (*p* = 0.0001), Young’s modulus (*p* < 0.0001), and yield strength (*p* = 0.0095), were observed (Figure 6). Based on obtained results, treatment one was selected (1 g L^−1^ TiO_2_ and pH 4.5) to perform photocatalytic transformation curves because the lowest contact angle and lowest weight at 300 h of UV light exposure were observed.

The treatment that presented the best results in terms of weight and SCA decrease was T1 pH 4.5 ± 0.2 and 1 gL^−1^ TiO_2_ (Figure 6). pH is a relevant photocatalytic factor as reported in the literature, since below the isoelectric point (pH of 6.5 ± 0.2) at acid conditions [6], TiO_2_ acquires a positive charge with *h+* generated as the dominant species with the highest oxidative capability, which allows attraction of compounds of the opposite charge (negatively charges), enabling their oxidation [48]. Therefore, plasma pretreatment negatively charged LDPE sheets because of forming polar groups on the surface. This in turn allowed for transitory adsorption phenomena to take place, generating the LDPE/TiO_2_ composite. These phenomena permitted more and better contact between the photo-exciter catalyzer and LDPE sheets, bringing about higher oxidation and photoinduced wear by the action of free radicals, and hence, photocatalytic degradation of the material. In the present work, the initial pH was 4.5 ± 0.2 and ended at 7.9 ± 0.2, an increase that could be attributed to (-OH) ion increase in the solution [6]. After exceeding the isoelectric point, TiO_2_ was a negative charge, thus separating from the material due to charge similarity. However, as observed in SEM images, cavity generation could have facilitated TiO_2_ particle capture, continuing with the photocatalytic degradation, which suggests that the photo-generated e^−^ are transferred to adsorbed O_2_, whose reduction allows the superoxide anion radical production (O_2_•^−^), which in turn favors hydroxyl radical formation (•OH) as dominant species [49], and this species was perhaps responsible for the degradation of the material.

LDPE photocatalytic degradation was evident from total organic carbon (TOC) and final weight data. Total carbon concentration increased from (4.07 ± 1.04) mg L^−1^ (at 0 h), to (6.73 ± 0.77) mg L^−1^ at 400 h, which could have been associated with fragmentation and hydrolysis of certain carbonated fractions of the polymer. Furthermore, the final weight decreased to 0.2 mg (Figure 7C).

TiO_2_ was a critical factor since high concentrations can minimize radiation reaching the material’s surface. Too high concentrations can decrease the effect UV can have on the system and photocatalytic degradation velocity, generating in the solution a “screen”-like effect [50].

Since the system operated with UV light it generated chemical photolysis, whose energy was sufficient to cleave the polymer’s covalent bonds, forming high-energy carbon-free radicals that reacted with O_2_ to form aldehydes and carboxylic acids [51], in addition to chain crosslinks. The photocatalytic effect was favored by low TiO_2_ concentrations in T1 treatment (Figure 6).

This work innovated in performing a photocatalysis with TiO_2_ in solution and the contaminant (LDPE sheets) in a solid phase, and not the other way around, as it is described for most reports in the literature [46,47]. This condition could have favored photocatalytic processes, since according to Bouna et al. [52], TiO_2_ adhered to a surface that can form films, which diminish the photocatalytic activity since the photocatalytic activity is less in a state of suspension [52]. For treatment of T1 during the first 100 h of the process, TiO_2_ particles were in an acid solution. According to Wang et al. [53], acid pH increases the number of aggregated particles, which drastically decreases TiO_2_ solubility [53]. Although this situation could have been present at the beginning of the process, it was minimized by the constant use of aeration within the photoreactor.

As in this work, others researchers have published that a mixture of the anatase–rutile phases, (Figure 5), present a greater yield or activity in comparison to using the anatase phase [54,55]. Results have been attributed to a higher redox potential due to electron transfer from the anatase conduction band to the rutile phase conduction band [56].

#### 2.4.3. Photocatalytic Transformation Curve at 400 h

While the process of photocatalysis took place, a decrease in SCA was observed, with a final value of (16 ± 3)°, which after 400 h of treatment represented an 82% reduction in comparison with pristine LDPE. For Young’s modulus, an increase was observed after the first 100 h, which gradually decreased to end in (47 ± 5) Mpa, representing a 38% increase (Figure 7A). Yield strength and pH behavior are presented in Figure 7B. As the process developed in time, LDPE sheets were modified and yield strength decreased with a final value of (9 ± 2) Mpa. In comparison with pristine LDPE value, a 10% decrease was observed. Concerning pH, the initial value was 4.5 ± 0.2. After 400 h of treatment, it ended in 7.9 ± 0.1 (Figure 7B). Furthermore, LDPE sheets presented a slight decrease in weight, which indicated the process of photocatalytic degradation was taking place (Figure 7C). Last, for TOC as was observed for SCA treatment, a sharp increase was observed during the first 100 h, with a gradual decrease (Figure 7C). Overall, TOC increased from 4.07 ± 1.04) mg L^−1^ (at 0 h), to (6.73 ± 0.77) mg L^−1^ at 400 h.

For SCA photolysis control, values were (61 ± 9)°, a value greater than treatments, a final weight of (4.2 ± 0.3) mg, like T4, and Young’s modulus of (48 ± 4) MPa, and yield strength of (4 ± 1) MPa, comparable with T2. Collectively, these data indicate photolysis by itself is not sufficient to generate a hydrophilic event on LDPE sheets and facilitate their degradation, yet in combination with plasma discharge, they can significantly increase the transformation effect.

#### 2.4.4. Fourier-Transform Infrared Spectroscopy (FTIR)

FTIR spectra for pristine LDPE (black line), O_2_ plasma-treated (red line), and sequential plasma followed by photocatalysis treatment (T1, P + P, blue line) appear in Figure 8A. LDPE distinctive signals (2920 cm^−1^, CH_2_ asymmetric stretching; 2851 cm^−1^, CH_2_ symmetric stretching; 1471 cm^−1^, CH_2_ bending and deformation; and 719 cm^−1^, CH_2_ rocking and deformation, were also present in spectra with plasma treatment (P + F), although with less intensity (Figure 8B–D)). Signals observed in pristine material and post-plasma material increased their signal after photocatalysis, at 610 cm^−1^ as a signal associated with the C-OH bond (twisting deformation) (Figure 8E). Moreover, signals at 1186 cm^−1^ and 1000 cm^−1^ (Figure 8G) were related to quaternary carbon and C-OH stretching bond, respectively. After plasma and photocatalysis, sequential treatments observed LDPE changes appear in Figure 8. As aforementioned for Figure 5, characteristic LDPE signal decrease at 2920 cm^−1^ (Figure 8B), could be attributed to ablation processes [27], which release carbon-containing material, decreasing LDPE weight [28]. During plasma discharge followed by photocatalysis (P + P), presence of two signals was observed (1200 cm^−1^ and between 1600–1800 cm^−1^) (Figure 8F), which corresponds to C-O vibrations and carbonyl signal (C=O), respectively. These later disappeared after 400 h of the photocatalytic process. In this latter one, an ample signal was observed in the vicinity of 610 cm^−1^ (Figure 8E), corresponding to deformation in the C-OH bond [45]. The presence of (C=O) groups during plasma discharge [23] and hydroxyl (OH-) during photocatalysis suggests the existence of LDPE oxidation processes for both treatments.

LDPE SEM images obtained from the 2^2^ factorial design are depicted in Figure 9, without analysis (9A) and analyzed through image processing (9B). It can be observed that treatment T1 (pH 4.5 ± 0.2 and 1 gL^−1^ TiO_2_) (Figure 9B,E) resulted in the highest surface modification (cavity appearance) of 8.86% (Figure 9B’) and 11.94% (Figure 9E’) for the same treatment at 400 h. Table 2 shows the response variables during sequential plasma-photocatalysis.

Scanning electron microscopy revealed LDPE’s superficial modifications after different photocatalytic treatments (Figure 9). Substantial topographical changes were observed when comparing pristine material (Figure 4A) with treated LDPE or even with LDPE submitted only to photolysis (Figure 8A). Out of the factorial design treatments, treatment T1 (pH 4.5 ± 0.2/1 gL^−1^ TiO_2_) (Figure 8B) presented the greatest surface changes, which were superior in the 400-h curve (Figure 8E), displaying cavities and a very rough surface. White particles on the surface corresponded to TiO_2_. The presence of cavities resulted from volatile product release from the polymeric matrix [57], due to TiO_2_/LDPE adsorption processes and later oxide reduction reactions during photocatalysis. These types of alterations have also been observed by other authors [57,58,59].

When comparing LDPE’s physical properties after the sequential plasma-photocatalysis processes with pristine material, it was observed that treatment generated a complementary effect that maintained hydrophilicity, attaining an 81.6% decrease for SCA. Another important change observed was a 38% increase in Young’s modulus, demonstrating these processes increased the material’s rigidity by restricting atom movement by the incorporation of polar groups, by crosslinking and bond cleavage in LDPE amorphous region, evidenced as cavities on the surface, which increase the crystalline zone proportion [29].

## 3. Materials and Methods

### 3.1. LDPE Sheets

LDPE sheets were purchased from a local market in Bogotá, D.C., Colombia. They were prepared for experiments as described [29].

### 3.2. Direct-Current Low-Pressure Plasma (DC-LLP) and Plasma Discharge

Within a high vacuum chamber (18 cm × 18 cm) with glass walls and stainless-steel lids and flanges, two flat electrodes, anode, and cathode, were placed with parallel cylinders separated by varying “s” distances to generate a low-temperature self-sustained DC plasma. Argon and oxygen reactive gas mix were rarefied at different concentrations. (Appendix A). Plasma discharge condition selection that favored LDPE sheet modification included: gas effect (Ar, O_2_, and a mix of Ar–O_2_ at a 1:1 ratio), voltage (V), and plasma chamber pressure (mbar). Selection of operating conditions and their effect on LDPE are detailed in Appendix A.

Once treatments that met these conditions were identified, operation settings that would decrease static contact angle (SCA) in LDPE sheets were selected as follows: 5treatments for O_2_, 13 for Ar, and 14 for the Ar–O_2_ mix (Appendix A). Differences among conditions for SCA-obtained results were analyzed, as well as mean comparison among treatments for each gas and mix (independent blocks), where the most significant for each one was selected (*p* < 0.05). The analysis was performed by ANOVA employing SAS^®^ software (SAS Institute 2017, version STAT 14.3. Cary, NC, USA: SAS Institute). Following, a new statistical analysis was performed for each block using the same tests previously selected (O_2_, Ar, and Ar–O_2_ mix). With the obtained results, subsequent experimental conditions were established [17,33,43,60].

### 3.3. Ablation Curve and Hydrophobic Recovery

Employing the glow discharge treatment that generated the lowest SCA, a new assay was performed in the wet chamber [29] to determine the effect plasma treatment would have on LDPE’s hydrophobicity changes as a function of time (ablation curve) [2]. To this end, five LDPE sheets were submitted to O_2_ plasma discharge for 30, 60, 120, and 240 s of exposure. As control, pristine LDPE sheets were used and results were expressed as final SCA in degrees ± standard deviation.

Additionally, following Mortazavi and Nosonovsky [12], LDPE sheet hydrophobic recovery [2] was evaluated for seven days. To this end, a new set of 10 pristine LDPE sheets was divided into two lots. The first five sheets were treated with plasma discharge and the remaining five were used as controls. To determine hydrophobic recovery percentage after ablation treatment, LDPE sheets were placed within a plastic box and left at room temperature (14° C), and for seven days SCA was measured [12].

### 3.4. TiO_2_/UV Photocatalysis

For photocatalysis, titanium dioxide (TiO_2_ Aldrich™) mixture of rutile and anatase (99.5% purity) was employed. To observe morphological characteristics and semi-quantitative composition, scanning electron microscopy images were captured, coupled with energy dispersion spectroscopy (SEM/EDS). A Jeol™ (Tescasn Brno, Czech Rep.) JSM 6490LV scanning electron microscope with 30kV, SEI signal, and 5000–1000× magnification was used. Observations were performed by depositing 0.5 g TiO_2_ on a glass substrate [61]. The sample was oven-dried at 50 °C for 24 h. To determine TiO_2_ crystal structure, X-ray diffraction was performed (XRD) using an X-Ray diffractometer Siemens D-5000 (Munich, Germany) equipment operated with a Cu Kα = 1.5418 Å anode [47,61,62].

For the photocatalysis experiments, a 2.5 L photoreactor (Sorvirel™-Bogotá, Colombia) with a work effective volume (WEV) of 1.2 L was employed. A 15 W UV lamp (Phillips™) with UV-C (280–100 nm) light emission capacity located within the photo-reactor and protected with a quartz glass jacket was employed. Within the reactor’s lower part, three ports for air injection were located and the air was injected at 10 ft^3^/h (SCFH) (4.7 L min^−1^). Positioned within the reactor and in parallel with the UV lamp, inert clamps were installed to hold previously O_2_ plasma-irradiated LDPE sheets (14 sheets of 3 cm^2^) [29] (Figure 10).

Condition selection for LDPE sheet photocatalytic transformation was performed through a 2^2^ factorial design where evaluated factors included Factor A: pH with a (−1) level of 4.5 ± 0.2 and a (+1) level of 9.0 ± 0.2. Factor B was TiO_2_ in gL^−1^, with a (−1) level of 1.0 and a (+1) level of 10 gL^−1^ (TiO_2_ was added within the reactor and homogenized by air injection with UV lamps off).

The design generated four codified treatments as follows: T1: −1 − 1 (pH: 4.5, TiO_2_ 1 gL^−1^), T2: +1 − 1 (pH: 9.0, TiO_2_ 1 gL^−1^), T3: −1 + 1 (pH: 4.5, TiO_2_ 10 gL^−1^), and T4: +1 + 1 (pH: 5, TiO_2_ 1 gL^−1^), performed in triplicate during 300 h of photocatalysis. After each treatment, LDPE sheets were removed to analyze response variables: SCA, final sheet weight, Young’s modulus, and yield strength [29]. Significant treatment selection was determined by an analysis of variance and a mean comparison using Design-Expert^®^ (Stat-Ease Inc. 2017 software, version 11.0. Minneapolis, MN: Stat-Ease) and SAS^®^ (SAS Institute 2017, version STAT 14.3. Cary, NC, USA: SAS Institute) [63].

The selected 2^2^ factorial design treatment was employed to perform photocatalytic degradation curves during 400 h; every 100 h, three LDPE sheets were removed to analyze changes associated with final weight (mg), SCA (°), Young’s modulus (Mpa), and yield strength (Mpa). Additionally, SEM and Fourier-transform infrared spectroscopy (FTIR) studies were performed. As controls, LDPE sheets exposed only to photolysis and plasma discharge were employed. Determinations for controls were performed at the beginning and the end of the process. Moreover, to determine total organic carbon (TOC) [63], pH aqueous solutions containing TiO_2_ were sampled, separating the chemical compound by sedimentation. Results obtained at 400 h of treatment were analyzed by ANOVA to determine differences among treatments. Additionally, multiple variable correlations with a confidence interval of 95% were performed using SAS^®^ software (SAS Institute 2017, version STAT 14.3. Cary, NC, USA: SAS Institute) [63].

### 3.5. Variables and Analytical Techniques Associated with LDPE Sheet Changes

#### 3.5.1. Static Contact Angle

To obtain LDPE’s surface microscopic information (atomic and/or molecular) in terms of its macroscopic properties, a correlation between static contact angle and tension or surface adhesion was established (hydrophilicity or wettability) [64,65] (Appendix A).

#### 3.5.2. Viscoelastic Properties

LDPE sheet Young´s modulus [39] and yield strength [66] were determined from stress-strain curves employing a Cobra 4 dynamometer, a THORLAB™ (Newton, NJ, USA) MTS25-Z8 motorized translation stage and Phywe Measure 4 (version 1.4, 2010) and ATP User (version 1.0.28, 2008) software for supervision. Control operating conditions were as follows: 0.1 mm s^−1^ testing speed, 250% maximum strain and 5 N maximum stress.

#### 3.5.3. Microscopy

Scanning electron microscopy (SEM) [67,68] (Jeol™ JSM 6490LV) with a 10 kV to 20 kV potency SEI signal and 500 and 6500× magnification was used to characterize LDPE’s surface. Samples were coated with gold in the Denton Vacuum Desk IV preparation system. Service was contracted with Universidad de Los Andes (UNIANDES), Bogotá, D.C., Colombia.

Atomic force microscopy (AFM) [30,64] [(Nanosurf ™ easy scan 2) Liestal, Switzerland] was used to study the surface´s changes in roughness. Parameters were: Mode: contact, Size: 61.8 µm, Setpoint: 20 nN; P-Gain: 1000; I-Gain: 100; D-Gain: 0. For roughness calculation, three measurements at different locations of the sample were carried out and mean ± SD was determined according to (Equation (1)) [69].
(1)$$R\;=\;\frac{1}{N}\;\overset{N}{\underset{i=1}{\sum }}\left|Zi\;-\;\bar{Z}\right|$$
where: *N* is surface height data number and $\bar{Z}$ mean height distance

#### 3.5.4. Computational Image Processing

Computational SEM image processing techniques were used to measure surface cavities produced by sequential treatment (plasma + TiO_2_/UV photocatalysis). All images were formatted to 1280 × 860 pixels and their noise was reduced by a Gaussian filter aided by a Euclidean curve shortening flow process to enhance edges [70]. The brightest sections were selected using the morphological Top-Hat transformation [71]. Lastly, a binary process based on morphology was used to select pixels, whose intensity was above a base threshold. The cavity area was calculated as the number of pixels selected divided by the total number of pixels in the image.

#### 3.5.5. Fourier-Transform Infrared Spectroscopy (FTIR)

Fourier-transform infrared spectroscopy was used to evaluate LDPE chemical bond modifications and chemical group composition [42,64,67]. A Shimadzu™ (Kyoto, Japan) IR Prestige-21 spectrophotometer was used by following the parameters reported by Gómez-Méndez et al. [29].

#### 3.5.6. Total Organic Carbon (TOC) and pH for the TiO_2_ Solution

For each sampling interval, 15 mL TiO_2_ solution was collected, and pH was measured in duplicate [63] using an Okaton™ (Skokie, IL, USA) pH meter. The sample could sediment for 24 h, 10 mL of the supernatant was collected and analyzed in triplicate with a TOC analyzer [63] (Shimadzu™ TOC-L).

## 4. Conclusions

LDPE sheets treated with 100% O_2_ plasma at 3.0 × 10^−2^ mbar, 600 V, six minutes of exposition showed in comparison with pristine LDPE a 75.58% reduction in SCA because of polar group incorporation (peroxides and alcohols) on the material’s surface, making the material hydrophilic, and a 94.89% increase in roughness, due to ablation and an increase in rigidity because of carbonated chain relative movement decrease from crosslinking and oxide reduction events. These oxidations facilitated the photocatalytic activity of a TiO_2_ solution (1 gL^−1^, pH 4.5 ± 0.2, 400 h), generating cavities on LDPE’s surface or expanding the ablation initiated by the plasma discharge, yet retaining high hydrophilicity and material’s rigidity, hence demonstrating a complementary effect of both processes. To the best of our knowledge, this is a pioneering work employing LDPE sheets in a sequential O_2_ plasma and TiO_2_/UV photocatalysis in solution, maintaining low SCA. Our data suggest sequential treatment with physicochemical processes in a complementary manner should be evaluated on plastic transformation.

## Figures and Tables

**Figure 1 molecules-26-02513-f001:**
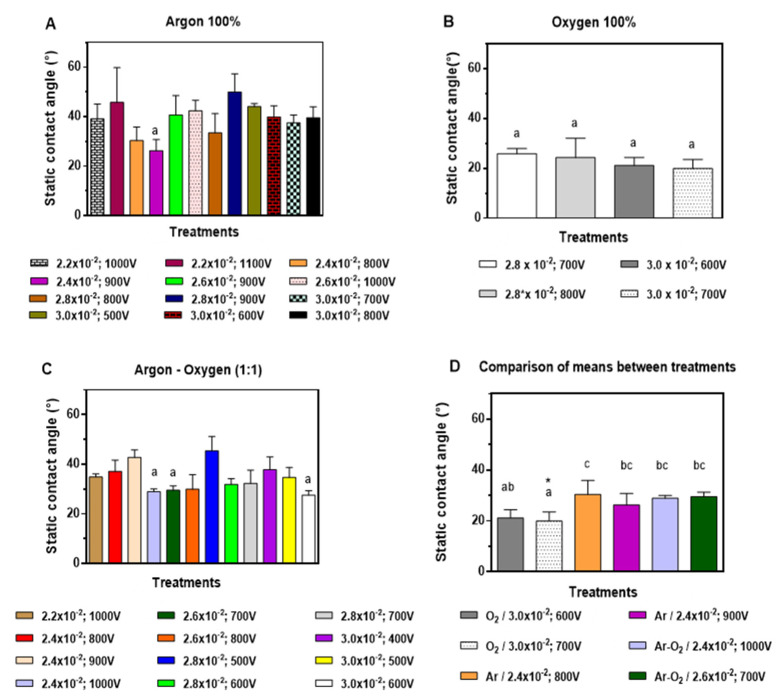
Static contact angle (°) of low-density polyethylene (LDPE sheets) exposed to: (**A**) 100% argon (*v*/*v*); (**B**) 100% oxygen (*v*/*v*), (**C**) 50:50% argon and oxygen (*v*/*v*). (**D**) Best treatments (mean ± SD, n = 5). Letters correspond to heterogeneous groups obtained by Tukey post hoc test, where letter a corresponds to the best treatment, followed by letters ab and bc. * is to higlight the best treatment.

**Figure 2 molecules-26-02513-f002:**
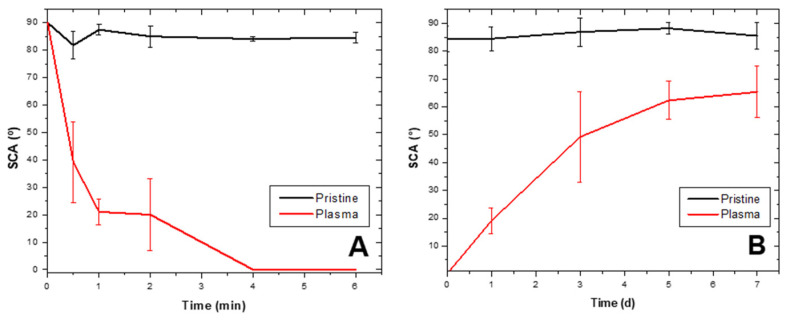
LDPE static contact angle changes after plasma treatment and recovery. (**A**) LDPE sheet static contact angle (SCA) decrease after 6-min treatment with O_2_ plasma discharge (ablation curve). (**B**) Hydrophobic recovery in LDPE sheets after being submitted to O_2_ plasma discharge and stored for 7 days, represented as changes in SCA.

**Figure 3 molecules-26-02513-f003:**
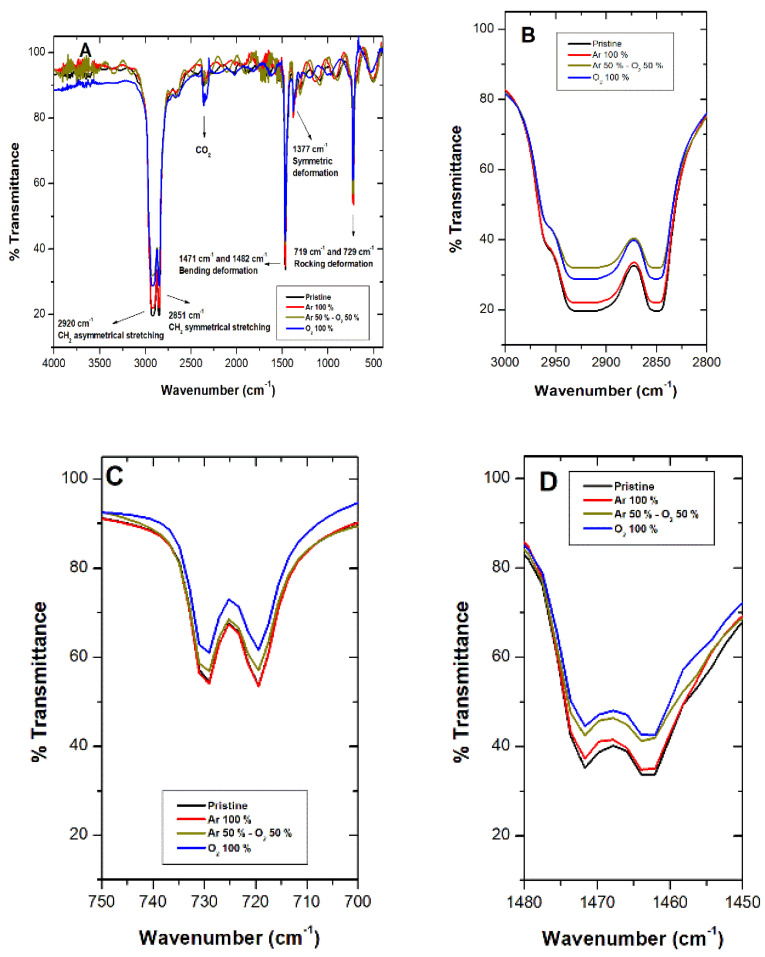
Fourier-transform infrared spectroscopy (FTIR) spectroscopy for LDPE sheets after plasma treatments with argon (Ar) (red line), oxygen (O_2_) (blue line), and Ar–O_2_ mix (brown line). The black line corresponds to LDPE pristine. (**A**) Complete spectra show LDPE characteristic signals. (**B**–**E**) Details characteristic of LDPE signals after plasma treatments.

**Figure 4 molecules-26-02513-f004:**
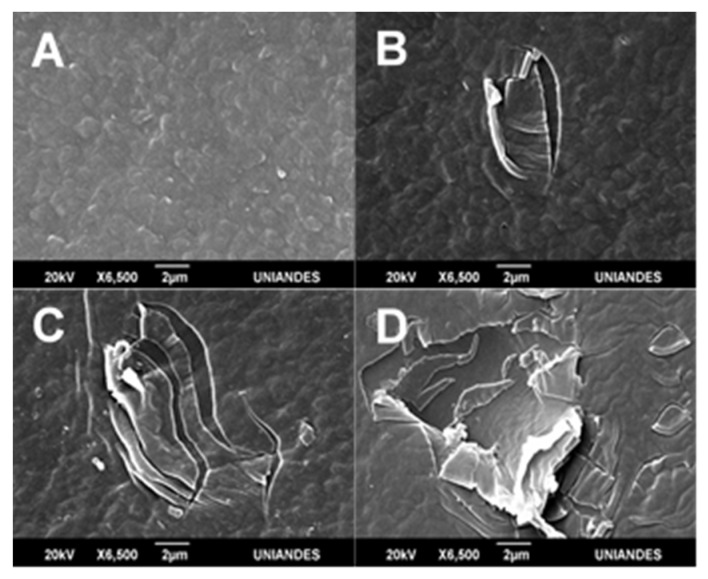
LDPE scanning electron microscopy (SEM) images submitted to Ar and O_2_ plasma treatments. Pristine LDPE (**A**). LDPE submitted to Ar plasma (**B**). Ar–O_2_ (1:1) mix (**C**). O_2_ plasma treatment (**D**).

**Figure 5 molecules-26-02513-f005:**
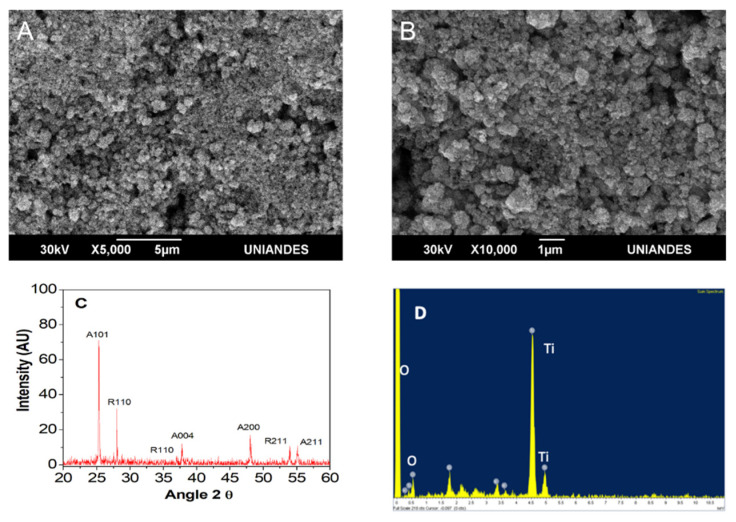
TiO_2_ (Aldrich) SEM and XRD analysis. (**A**) TiO_2_ (Aldrich) SEM images at 5000× TiO_2_ (Aldrich). (**B**) SEM at 10,000×. (**C**) EDS analysis. (**D**) XRD results.

**Figure 6 molecules-26-02513-f006:**
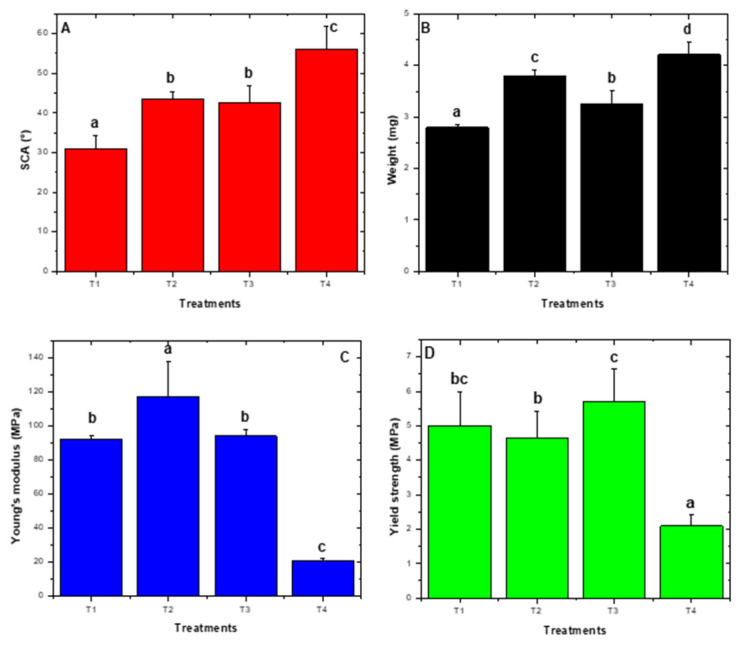
2^2^ Factorial design response variables. (**A**) Static contact angle. (**B**) weight. (**C**) Young’s modulus. (**D**) Yield strength. For SCA (35 ± 5)°, T1 was significant whereas for weight, T1 (3.4 ± 0.5) mg and T3 (3.6 ± 0.6) mg were significant. For Young´s modulus, T4 was significant (21 ± 1) MPa and yield strength (2 ± 1) MPa. T1: pH: 4.5, TiO_2_ 1 gL^−1^, T2: pH: 9.0, TiO_2_ 1 gL^−1^, T3: pH: 4.5, TiO_2_ 10 gL^−1^, and T4 pH: 9.0, TiO_2_ 10 gL^−1^. Letters a, b, c, d correspond to heterogeneous groups.

**Figure 7 molecules-26-02513-f007:**
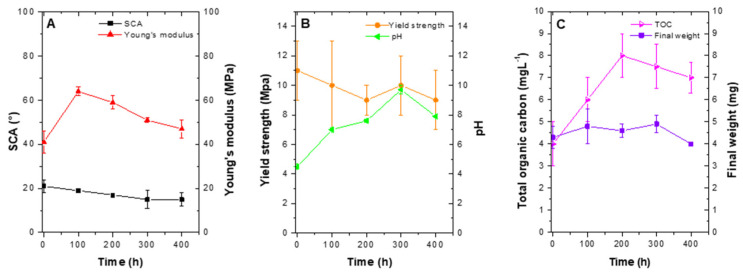
Photocatalytic transformation curves after 400 h of treatment. (**A**) SCA and Young´s modulus. (**B**) Yield strength and pH y. (**C**) Total organic carbon and final weight. (Treatment conditions: 1 g L^−1^ TiO_2_ and pH 4.5).

**Figure 8 molecules-26-02513-f008:**
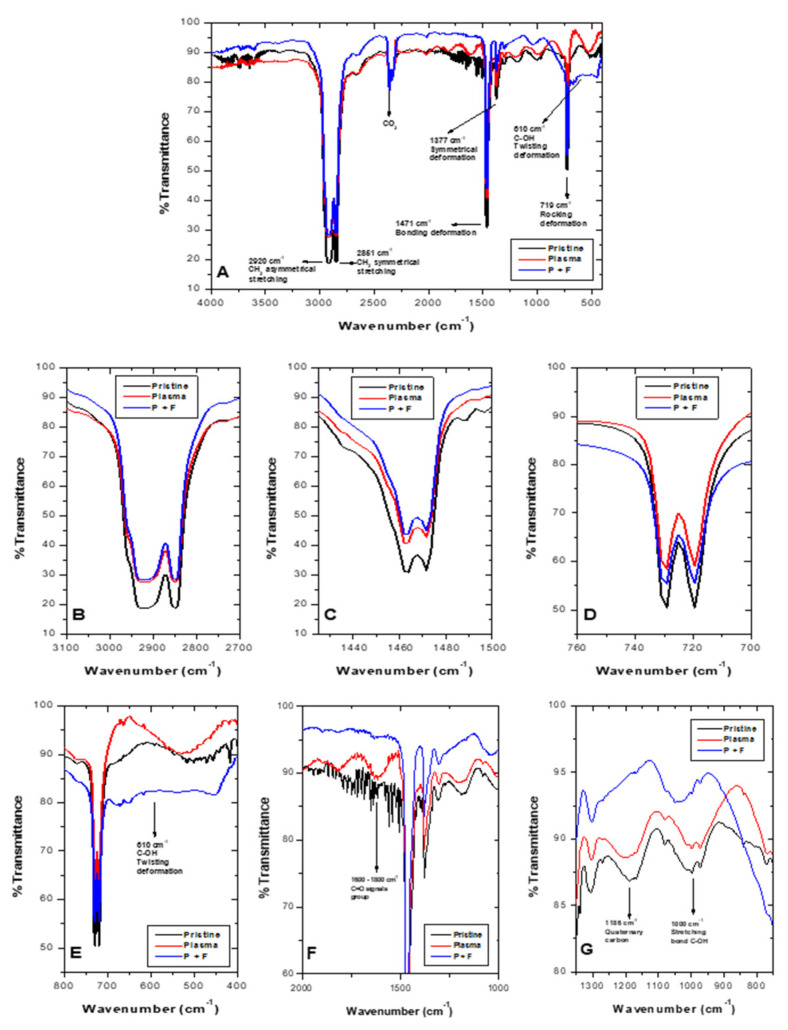
FTIR spectra for untreated and treated LDPE sheets. Pristine LDPE (black line), post-plasma (red line), and after sequential plasma and photocatalysis (blue line). (**A**–**D**) Regions characteristic of LDPE, where after the treatments, a decrease in signal appeared. (**E**–**G**) Regions where changes after the sequential process occurred.

**Figure 9 molecules-26-02513-f009:**
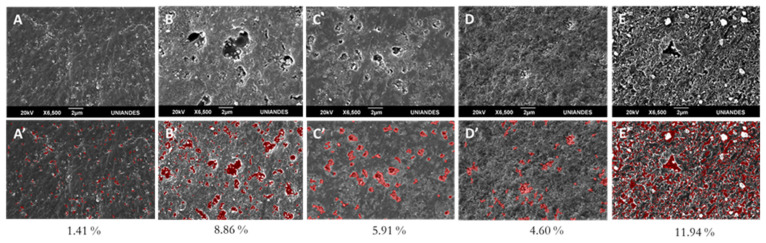
LDPE SEM images after different photocatalytic degradation treatments. (**A**–**E**) LDPE SEM images after different photocatalytic degradation treatments (2^2^ factorial design for 400 h) without analysis. (**A**′–**E**′) Same treatments with image analysis. In red, percentage quantification of generated cavities for different treatments. (**A**) Photolysis control. (**B**) T1. (**C**) T2. (**D**) T4 and (**E**) T4 at 400 h.

**Figure 10 molecules-26-02513-f010:**
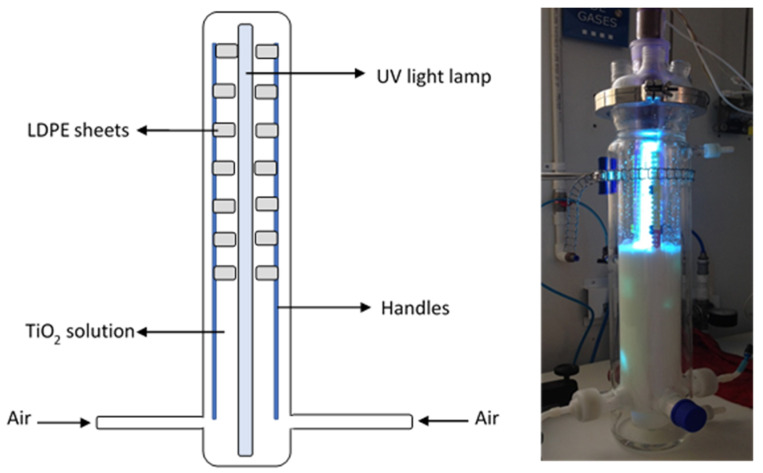
Photoreactor’s assembly for the photocatalysis process.

**Table 1 molecules-26-02513-t001:** Pristine low-density polyethylene (LDPE) sheet initial characteristics.

Response Variables	Value	References
Weight (mg)	4.3 ± 0.5	Present work
Static contact angle (°)	87 ± 1	Present work
Roughness (nm)	6 ± 2	Present work
Young’s modulus (MPa)	34 ± 1	Present work
Yield strength (MPa)	10 ± 2	Present work
Spectra and vibrational band structure (FTIR)	729 cm^−1^ CH_2_ rocking and deformation	[27,28]
	1460 cm^−1^ CH_2_ bending and deformation	[27,28]
	2921 cm^−1^ CH_2_ asymmetric stretching	[27,28]
	2843 cm^−1^ CH_2_ symmetric stretching	[27,28]

**Table 2 molecules-26-02513-t002:** Comparison of response variables during sequential plasma-photocatalysis treatment.

Response Variables	Pristine	Plasma	Photocatalysis
Weight (mg)	4.3 ± 0.5	4.2 ± 0.1	4.0 ± 0.1
Static contact angle (°)	87 ± 1	21 ± 3	16 ± 3
Young’s modulus (MPa)	34 ± 1	41 ± 6	46 ± 5
Yield strength (MPa)	10 ± 2	11 ± 2	9 ± 2

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
