# Peer review of "LDPE Transformation by Exposure to Sequential Low-Pressure Plasma and TiO2/UV Photocatalysis"

_molecules, 2021, doi:10.3390/molecules26092513_

Round 1

Reviewer 1 Report

Dear Authors,

The manuscript “LDPE transformation by exposure to sequential low-pressure plasma and TiO2/UV photocatalysis” by L. D. Gomez-Mendez et al. is devoted to the study of mechanical and surface properties of LDPE after these plasma and photocatalytic treatments. The topic is quite interesting since plastic pollution is a crucial worldwide issue, especially for the effect on the natural environment. In addition, the introduction is well written which is a positive point.

However, it cannot be published in the present form since the discussion part is not complete. Indeed, you discussed and compared too much with the existing literature and not comprehensively detailed and discussed your results. It is a research article, not a review article. In addition, the manuscript is too long and needs to be shortened. For example, I suggest to merge the result and discussion sections in one, so it will be more concise. I strongly suggest to perform an extensive English language editing by a native speaker (the manuscript is not easy to understand, and one reason is the quality of English language). Furthermore, section numbering and figure numbering is not consistent (for example, the first figures is Fig.4!).

Hereafter are a non-exhaustive list of remarks/comments that have to be addressed:

  • Abstract: “DC electric glow discharge” should be followed by “i.e. low-pressure plasma” since it should be consistent with the title of the work.
  • Abstract: you mention often the SCA and it appears a crucial factor. You should explain why SCA is so important.
  • Introduction – line 51-57: I suggest to insert a scheme to explain all the processes that occur during abalation (because you described a lot of reactions).
  • Introduction – line 71: I suggest to delete “or composites based on plastic or oxide semiconductor”.
  • Introduction – line 73: please insert the greek letter for the wavelength before “<400”.
  • Introduction – lines 79-80: the photoexcited electron are in CB (not VB) and photogenerated holes are in VB. Both are responsible of formation of ROS such as HO* and O2*-.
  • Introduction – line 89: please insert references for “has been less studied”.
  • Results (and discussion): as I already mentioned, there is too many subsections. I suggest organizing as follows: 2 sections on mechanical and surface properties for each treatment (i.e. plasma and photocatalysis). It would be better to read. In addition, merge the result and discussion in one section.
  • Results – Table 1: what is reference “3”?
  • Results – line 120: Briefly explain why lower voltage is chosen (although it is obvious).
  • Results – lines 148-151: The sentence is not clear. You compared the values of treated LDPE with what? Because you mentioned significant changes but we do not know compared to what. Later you mentioned there is not significant change compared to pristine LDPE. Please, clarify.
  • Results – line 210: you observed particle size in the range of nm in the SEM? How it is possible? I can only see that the size is in the range of µm. Please, correct it.
  • Results – lines 215-219: You mentioned in EDS that there is 3 at% of Ti and 96 at% of O. So it is not TiO2! Please, clarify.
  • Results – line 221-226: it is necessary to analyze by XRD the exact phase composition of the powder. Or if you are using commercial powders, it is useless to make such analyses since it is provide by the supplier...
  • Results – line 238: TiO2 is considered as a factor. What do you mean? Concentration?
  • Discussion – line 327: “in their works” is not correct since it is yours. Please, correct it.
  • Discussion – line 338: in this paragraph, you starts explaining you used Ar+O2 gases and you discussed why it is good (by supporting with some references), but then you explained that O2 is the best condition. You should rearrange the paragraph or better make the link with the next paragraph since it is explaining it. For example, I would start the next paragraph by “The difference between Ar and O2 discharge are that...”.
  • Discussion – line 342-354: in this paragraph, you mostly discussed the literature, and not your results. That is one reason you should merge the result and discussion sections in one.
  • Discussion – line 447: you explained that the positive charge surface is acquired with photogenerated h+. So what about the photogenerated e-? Do you mean h+ migrate at the surface and e- toward the bulk? If yes, you should prove it. By the way, I think it is better to discuss the charge surface using pH and isoelectric point.
  • Discussion – line 462: you do not discuss your results. You mentioned HO* and O2*- radicals, so you have to prove their formation by your own experimental results.
  • Discussion – lines 476-479: you used TiO2 as photocatalyst under UVC light. So you have mainly photolysis than photocatalysis. You should do all your experiments with UVC treatment only without TiO2 since you should determine the contribution of intrinsic photocatalysis and photolysis. If you want to make TiO2 photocatalysis, you have to use UVA or UVB. So please, do it.
  • Discussion – lines 499-509: the discussion is interesting but please, explain the plausible mechanisms.
  • Materials and Methods: this section is too long. It should be shortened and/or placed in supplementary information.

Author Response

Firstly, the authors thank reviewers and Editor for their valuable comments and corrections. We are certain they will improve the quality of our work.

We answered every inquiry in red and manuscript modifications appear in red as well.

The manuscript “LDPE transformation by exposure to sequential low-pressure plasma and TiO2/UV photocatalysis” by L. D. Gomez-Mendez et al. is devoted to the study of mechanical and surface properties of LDPE after these plasma and photocatalytic treatments. The topic is quite interesting since plastic pollution is a crucial worldwide issue, especially for the effect on the natural environment. In addition, the introduction is well written which is a positive point.

However, it cannot be published in the present form since the discussion part is not complete. Indeed, you discussed and compared too much with the existing literature and not comprehensively detailed and discussed your results. It is a research article, not a review article.

In addition, the manuscript is too long and needs to be shortened. For example, I suggest to merge the result and discussion sections in one, so it will be more concise. I strongly suggest to perform an extensive English language editing by a native speaker (the manuscript is not easy to understand, and one reason is the quality of English language).

R/ We appreciate the referee's suggestion but, according to the journal's instructions for authors, the results and discussion sections should be separated. We will not merge the two sections. On the other hand, the manuscript was revised to summarize it.

Furthermore, section numbering and figure numbering is not consistent (for example, the first figures is Fig.4!).

R/ The numbering of figures was reorganized, considering that the journal first presents results, discussion and finally materials and methods.

Hereafter are a non-exhaustive list of remarks/comments that have to be addressed:

  • Abstract: “DC electric glow discharge” should be followed by “i.e. low-pressure plasma” since it should be consistent with the title of the work.

R/Done

  • Abstract: you mention often the SCA and it appears a crucial factor. You should explain why SCA is so important.

R/ It was made clear in the summary what this determination is made for.

  • Introduction – line 51-57: I suggest inserting a scheme to explain all the processes that occur during ablation (because you described a lot of reactions).

R/ We appreciate the referee's suggestion, but according to the journal's instructions and several articles we reviewed, the introduction does not include outlines. However, if the editor considers that this can be done, you could convert the text related to the ablation mechanisms into an outline. For now it is left as text.

  • Introduction – line 71: I suggest to delete “or composites based on plastic or oxide semiconductor”.

R/ Done

  • Introduction – line 73: please insert the greek letter for the wavelength before “<400”.

R/ Done

  • Introduction – lines 79-80: the photoexcited electron are in CB (not VB) and photogenerated holes are in VB. Both are responsible of formation of ROS such as HO* and O2*-.

R/ In the introduction it is said that when TiO2 is irradiated (xxx) a photoexcitation process takes place in which the energy of the photons is absorbed by the electrons in the Valencia band. But we are not saying that the photo-excited electrons are in the valence band. Subsequently, photogenerated electron vacancies or holes are generated in the valence band and the photo-excited electrons associate with the conduction band. The two form the electron (conduction band) and hole (Valencia band) pairs.

  • Introduction – line 89: please insert references for “has been less studied”.

R/Done

  • Results (and discussion): as I already mentioned, there is too many subsections. I suggest organizing as follows: 2 sections on mechanical and surface properties for each treatment (i.e. plasma and photocatalysis). It would be better to read. In addition, merge the result and discussion in one section.

R/ We thank the referee for the suggestion. The numbering is reorganized for the ablation and photocatalysis sections, but the subsections were not removed because for each treatment initial characterizations, selection of conditions (experimental designs) and finally transformation curves for both ablation and photocatalysis had to be done.

  • Results – Table 1: what is reference “3”?

R/ The number three was deleted

  • Results – line 120: Briefly explain why lower voltage is chosen (although it is obvious).

R/ A sentence was added to briefly explain why the low voltage was selected. Because the combination of these ablation conditions did not burn the foils, a stable plasma was obtained, and the contact ratio was decreased, therefore a hydrophilic surface was obtained compared to the pristine plastic which is hydrophobic.

Results – lines 148-151: The sentence is not clear. You compared the values of treated LDPE with what? Because you mentioned significant changes but we do not know compared to what. Later you mentioned there is not significant change compared to pristine LDPE. Please, clarify.

R/ Done

Results – line 210: you observed particle size in the range of nm in the SEM? How it is possible? I can only see that the size is in the range of µm. Please, correct it.

R/ Done

  • Results – lines 215-219: You mentioned in EDS that there is 3 at% of Ti and 96 at% of O. So it is not TiO2! Please, clarify.

R/ If it is TiO2 because it is reported 3.19 % Titanium, 96.01 % O2, 0.55 % Si and 0.25 % K, for a total of 100 %. Additionally, figure 5 shows the EDS result where the signal of O2 and Ti is appreciated, for the other elements the intensity of the signal was low and was not highlighted in the figure.

Results – line 221-226: it is necessary to analyze by XRD the exact phase composition of the powder. Or if you are using commercial powders, it is useless to make such analyses since it is provided by the supplier...

R/ As mentioned in the methodology and results, TiO2 from the commercial company Sigma-Aldrich was used, although they report the phases in the data sheet, it is necessary to confirm them through X-ray diffraction analysis. Because it is important to have the anatase phase and to observe which orientations are present.

Results – line 238: TiO2 is considered as a factor. What do you mean? Concentration?

R/ TiO2 expressed in g/L was the factor B evaluated in the experimental design. The corresponding unit was added

Discussion – line 327: “in their works” is not correct since it is yours. Please, correct it.

R/ Done

Discussion – line 338: in this paragraph, you starts explaining you used Ar+O2 gases and you discussed why it is good (by supporting with some references), but then you explained that O2 is the best condition. You should rearrange the paragraph or better make the link with the next paragraph since it is explaining it. For example, I would start the next paragraph by “The difference between Ar and O2 discharge are that...”.

R/ Done

Discussion – line 342-354: in this paragraph, you mostly discussed the literature, and not your results. That is one reason you should merge the result and discussion sections in one.

  • Discussion – line 447: you explained that the positive charge surface is acquired with photogenerated h+. So what about the photogenerated e-? Do you mean h+ migrate at the surface and e- toward the bulk? If yes, you should prove it. By the way, I think it is better to discuss the charge surface using pH and isoelectric point.

R/ This paragraph was rewritten to focus on pH and its relationship to the isoelectric point, which means that TiO2 at acidic pH can have a positive surface charge and plasma pre-treated plastic films have a negative surface charge.

Discussion – line 462: you do not discuss your results. You mentioned HO* and O2*- radicals, so you have to prove their formation by your own experimental results.

R/ It was not measured, but it was included as a possible mechanism and not as if we had tested it.

Discussion – lines 476-479: you used TiO2 as photocatalyst under UVC light. So you have mainly photolysis than photocatalysis. You should do all your experiments with UVC treatment only without TiO2 since you should determine the contribution of intrinsic photocatalysis and photolysis. If you want to make TiO2 photocatalysis, you have to use UVA or UVB. So please, do it.

R/ Done

Discussion – lines 499-509: the discussion is interesting but please, explain the plausible mechanisms.

Materials and Methods: this section is too long. It should be shortened and/or placed in supplementary information.

Reviewer 2 Report

The manuscript is devoted to the LDPE transformation by exposure to sequential low-pressure plasma and TiO2 catalyzed photooxidation. The manuscript contains good introduction, a lot of experimental results and discussion.

Nevertheless, a lot of remarks appears when reading the manuscript.

Please, decode the DC abbreviation. Do not use abbreviation in the title.

Abstract: Low-density polyethylene sheets (3.0 ± 0.1 cm) – 10 microns thickness! Describe the thickness, but not the area. 

Lines 89-90. Therefore, the objective of this study was to select reaction conditions and/or operating conditions to perform plasma discharge and TiO2/UV photocatalysis, as technologies to transform LDPE sheets. What for are the authors to transform LDPE sheets?

Lines 264-265. Figure 9: 22 Factorial design response variables (A) Static contact angle (B) weight. For SCA (35 ± 5) °, T1 was significant whereas for weight T1 (3.4 ± 0.5) mg and T3 (3.6 ± 0.6) mg, were significant. For Young´s modulus, T4 was significant (21 ± 1) MPa and yield strength (2 ± 1) MPa. I understood nothing, what are T1-T4? Such methods like 22 Factorial design do not produce scientific novelty and, hence, useless and  fruitless. Where is the answer on the main question for a catalytic reaction: how does the catalyst influence on the rate of the reaction? Why did not you make the experiment without TiO2? How much is the rate of photooxidation without the catalyst? What for is it necessary to apply 400-hours photooxidation after 4-minuts plasma oxidation? The answers on these questions should be given in the Conclusions.

Lines 289-290. Figure 10: Photocatalytic transformation curves after 400 h of treatment. (A) SCA and Young´s modulus. (B) Yield 289 strength and pH y (C) Total organic carbon and Final weight. Where are the conditions of the experiment?

Lines 455-456. In the present work, the initial pH was 4.5 ± 0.2 and ended at 7.9 ± 0.2, an increase that could be attributed to (-OH) ion increase in the solution or to bicarbonate  ion production (HCO3 - ) from CO2 gas, a volatile product of photocatalytic degradation  [6]. CO2 cannot increase pH. Oxidation of PE cannot increase pH of water phase. Hence, the pH increase may be connected with TiO2 impurities.

Lines 523 -529. When comparing LDPE’s physical properties after the sequential plasma-photocatalysis processes with pristine material, it was observed treatment generated a synergistic effect that maintained hydrophilicity attaining an 81.6 % decrease for SCA. Where are proves for a synergistic effect?  Another important change observed was a 38 % increase in Young’s modulus, demonstrating these processes increased the material’s rigidity by restricting atom movement by the incorporation of polar groups, by crosslinking and bond cleavage in LDPE amorphous region, evidenced as cavities on the surface, which increase the crystalline zone proportion [27]. All these improvements or “improvements” may be attained by more efficient methods (oxidation by Fenton reagent etc.) during minutes, but not 400 hours!

Several little confusions are shown below.

consecutive biodeterioration with P. ostreatus.- Italic! In their work, it was demonstrated plasma

generating carbon Alkyl radicals (C•) by splitting of C-H and C-C bonds. These radicals prefer 

the environment and form peroxide (C-O-O•) and hydroperoxide (C-O-OH) radicals  hydroperoxide (C-O-OH) is a molecule, but not radical

Author Response

Firstly, the authors thank reviewers and Editor for their valuable comments and corrections. We are certain they will improve the quality of our work.

We answered every inquiry in red and manuscript modifications appear in red as well.

The manuscript is devoted to the LDPE transformation by exposure to sequential low-pressure plasma and TiO2 catalyzed photooxidation. The manuscript contains good introduction, a lot of experimental results and discussion.

Nevertheless, a lot of remarks appears when reading the manuscript.

Please, decode the DC abbreviation. Do not use abbreviation in the title.

R/ Done

Abstract: Low-density polyethylene sheets (3.0 ± 0.1 cm) – 10 microns thickness! Describe the thickness, but not the area. 

R/ In the summary the area or dimensions of the sheets are presented. We are not presenting the thickness values.

Lines 89-90. Therefore, the objective of this study was to select reaction conditions and/or operating conditions to perform plasma discharge and TiO2/UV photocatalysis, as technologies to transform LDPE sheets. What for are the authors to transform LDPE sheets?

R/ Done

Lines 264-265. Figure 9: 22 Factorial design response variables (A) Static contact angle (B) weight. For SCA (35 ± 5) °, T1 was significant whereas for weight T1 (3.4 ± 0.5) mg and T3 (3.6 ± 0.6) mg, were significant. For Young´s modulus, T4 was significant (21 ± 1) MPa and yield strength (2 ± 1) MPa. I understood nothing, what are T1-T4? Such methods like 22 Factorial design do not produce scientific novelty and, hence, useless and fruitless. Where is the answer on the main question for a catalytic reaction: how does the catalyst influence on the rate of the reaction? Why did not you make the experiment without TiO2? How much is the rate of photooxidation without the catalyst? What for is it necessary to apply 400-hours photooxidation after 4-minuts plasma oxidation? The answers on these questions should be given in the Conclusions.

 R/ All questions are answered separately below.

  • What are T1-T4?Such methods like 22 Factorial design do not produce scientific novelty and, hence, useless and fruitless.

 Response: A 22 factorial design evaluates the effect of two factors at two levels on a given response variable. T1 to T4 are the factors that generated the design and are presented in the methodology as 1 -1, +1-1, -1+1, and +1+1. To facilitate the understanding of the design, in the methodology section and legend of figure 6, we add what each of the four treatments consisted of. On the other hand, experimental designs are carried out to determine the effect of factors and are helpful because they help to reduce the number of experiments. They are relevant to this article because pH and catalyst concentration are conditions that influence the photocatalysis process and the ones that favour the process should be selected.

  • Where is the answer on the main question for a catalytic reaction: how does the catalyst influence on the rate of the reaction?

R/ The response variables of the factorial design were measured as a function of a fixed time and not as reaction kinetics. The aim of the experiment was to select the levels of the two factors (pH and TiO2 concentration) that favour the transformation process of the plastic films. Therefore, no kinetics were performed to calculate the reaction rate by varying the catalyst concentration.

Why did not you make the experiment without TiO2? How much is the rate of photooxidation without the catalyst?

R/Done

  • What for is it necessary to apply 400-hours photooxidation after 4-minuts plasma oxidation? The answers on these questions should be given in the Conclusions

R/ Done

Lines 289-290. Figure 10: Photocatalytic transformation curves after 400 h of treatment. (A) SCA and Young´s modulus. (B) Yield 289 strength and pH y (C) Total organic carbon and Final weight. Where are the conditions of the experiment?

R/Done

Lines 455-456. In the present work, the initial pH was 4.5 ± 0.2 and ended at 7.9 ± 0.2, an increase that could be attributed to (-OH) ion increase in the solution or to bicarbonate ion production (HCO3 -) from CO2 gas, a volatile product of photocatalytic degradation [6]. CO2 cannot increase pH. Oxidation of PE cannot increase pH of water phase. Hence, the pH increase may be connected with TiO2 impurities.

R/ Done

Lines 523 -529. When comparing LDPE’s physical properties after the sequential plasma-photocatalysis processes with pristine material, it was observed treatment generated a synergistic effect that maintained hydrophilicity attaining an 81.6 % decrease for SCA. Where are proves for a synergistic effect?  Another important change observed was a 38 % increase in Young’s modulus, demonstrating these processes increased the material’s rigidity by restricting atom movement by the incorporation of polar groups, by crosslinking and bond cleavage in LDPE amorphous region, evidenced as cavities on the surface, which increase the crystalline zone proportion [27]. All these improvements or “improvements” may be attained by more efficient methods (oxidation by Fenton reagent etc.) during minutes, but not 400 hours!

R/Done

Several little confusions are shown below.

consecutive biodeterioration with P. ostreatus- Italic! In their work, it was demonstrated plasma

R/ The names of microorganisms are written in intalica. P. ostreautus is the correct spelling of Pleurotus (genus) and ostreatus (species).

generating carbon Alkyl radicals (C•) by splitting of C-H and C-C bonds. These radicals prefer 

R/Done

the environment and form peroxide (C-O-O•) and hydroperoxide (C-O-OH) radicals hydroperoxide (C-O-OH) is a molecule, but not radical

R/Done

Reviewer 3 Report

The manuscript titled "LDPE transformation by exposure to sequential low-pressure plasma and TiO2/UV photocatalysisLDPE transformation by exposure to sequential low-pressure plasma and TiO2/UV photocatalysis" presents an interesting study on sequential combined effect of plasma exposure and TiO2/UV photocatalysis on the properties of LDPE potentially beneficial for its degradation. Overall the, the study has some degree of novelty, experimental design is appropriate, analytical tools used are appropriate and the manuscript is written well.

There are  numerous typos that should be taken care of before it is published. Few examples have been

  1. Line: 24; "con-centration"
  2. Line: 97, Table 1. Webnumbers shoud be superscipted (cm-1 ) not as cm -1.

Author Response

Firstly, the authors thank reviewers and Editor for their valuable comments and corrections. We are certain they will improve the quality of our work.

We answered every inquiry in red and manuscript modifications appear in red as well.

The manuscript titled "LDPE transformation by exposure to sequential low-pressure plasma and TiO2/UV photocatalysis LDPE transformation by exposure to sequential low-pressure plasma and TiO2/UV photocatalysis" presents an interesting study on sequential combined effect of plasma exposure and TiO2/UV photocatalysis on the properties of LDPE potentially beneficial for its degradation. Overall the, the study has some degree of novelty, experimental design is appropriate, analytical tools used are appropriate and the manuscript is written well.

There are numerous typos that should be taken care of before it is published. Few examples have been

  1. Line: 24; "con-centration"

R/ Done

  1. Line: 97, Table 1. Webnumbers shoud be superscipted (cm-1 ) not as cm -1.

R/ Done

Round 2

Reviewer 1 Report

Dear authors,

The work was not improved. You did not take into considerations most of my comments and you have either proposed rebuttal answers or ignored the comments. Honestly, the revision was not prepared seriously.

I can only recommend the REJECTION of your article.

Regards,

C1/ The manuscript “LDPE transformation by exposure to sequential low-pressure plasma and TiO2/UV photocatalysis” by L. D. Gomez-Mendez et al. is devoted to the study of mechanical and surface properties of LDPE after these plasma and photocatalytic treatments. The topic is quite interesting since plastic pollution is a crucial worldwide issue, especially for the effect on the natural environment. In addition, the introduction is well written which is a positive point.

However, it cannot be published in the present form since the discussion part is not complete. Indeed, you discussed and compared too much with the existing literature and not comprehensively detailed and discussed your results. It is a research article, not a review article.

In addition, the manuscript is too long and needs to be shortened. For example, I suggest to merge the result and discussion sections in one, so it will be more concise. I strongly suggest to perform an extensive English language editing by a native speaker (the manuscript is not easy to understand, and one reason is the quality of English language).

R/ We appreciate the referee's suggestion but, according to the journal's instructions for authors, the results and discussion sections should be separated. We will not merge the two sections. On the other hand, the manuscript was revised to summarize it.

C2/ It is clearly written in the journal’s instructions that the Discussion part could be merged with the Results part! Then, no extensive english language editing was performed. Furthermore, the quality of the discussion was not significantly improved and it remains quite poor.

C1/ Introduction – line 51-57: I suggest inserting a scheme to explain all the processes that occur during ablation (because you described a lot of reactions).

R/ We appreciate the referee's suggestion, but according to the journal's instructions and several articles we reviewed, the introduction does not include outlines. However, if the editor considers that this can be done, you could convert the text related to the ablation mechanisms into an outline. For now it is left as text.

C2/ Please, you have to ask the editor. Indeed, many articles contain figures in the introduction. To insert a scheme in the form of a figure would be definitively good for the reader.

C1/ Introduction – lines 79-80: the photoexcited electron are in CB (not VB) and photogenerated holes are in VB. Both are responsible of formation of ROS such as HO* and O2*-.

R/ In the introduction it is said that when TiO2 is irradiated (xxx) a photoexcitation process takes place in which the energy of the photons is absorbed by the electrons in the Valencia band. But we are not saying that the photo-excited electrons are in the valence band. Subsequently, photogenerated electron vacancies or holes are generated in the valence band and the photo-excited electrons associate with the conduction band. The two form the electron (conduction band) and hole (Valencia band) pairs.

C2/ It is written: „photoexcited electrons in the VB are responsible for the formation of highly reactive oxidative species (reduced state), such as (OH*).“ It is not correct excpet in photoexcited electrons are photogenerated holes. The photogenerated e- in CB can form O2*- and the photogenerated h+ in VB can form HO*.

C1/ Introduction – line 89: please insert references for “has been less studied”.

R/Done

C2/ You changed the sentence for „has not been studied“. Be sure there is really no study combining these processes in any type of systems.

C1/ Results (and discussion): as I already mentioned, there is too many subsections. I suggest organizing as follows: 2 sections on mechanical and surface properties for each treatment (i.e. plasma and photocatalysis). It would be better to read. In addition, merge the result and discussion in one section.

R/ We thank the referee for the suggestion. The numbering is reorganized for the ablation and photocatalysis sections, but the subsections were not removed because for each treatment initial characterizations, selection of conditions (experimental designs) and finally transformation curves for both ablation and photocatalysis had to be done.

C2/ I do not agree. It can be reorganized.

C1/ Results – line 210: you observed particle size in the range of nm in the SEM? How it is possible? I can only see that the size is in the range of µm. Please, correct it.

R/ Done

C2/ Nothing was done!

C1/ Results – lines 215-219: You mentioned in EDS that there is 3 at% of Ti and 96 at% of O. So it is not TiO2! Please, clarify.

R/ If it is TiO2 because it is reported 3.19 % Titanium, 96.01 % O2, 0.55 % Si and 0.25 % K, for a total of 100 %. Additionally, figure 5 shows the EDS result where the signal of O2 and Ti is appreciated, for the other elements the intensity of the signal was low and was not highlighted in the figure.

C2/ The percentage are wt% or at%? My calculations do not show it can correlate with the formula TiO2.

C1/ Results – line 238: TiO2 is considered as a factor. What do you mean? Concentration?

R/ TiO2 expressed in g/L was the factor B evaluated in the experimental design. The corresponding unit was added

C2/ Ok, but you should explicitly write „TiO2 concentration“ and not the unit...

C1/ Discussion – line 338: in this paragraph, you starts explaining you used Ar+O2 gases and you discussed why it is good (by supporting with some references), but then you explained that O2 is the best condition. You should rearrange the paragraph or better make the link with the next paragraph since it is explaining it. For example, I would start the next paragraph by “The difference between Ar and O2 discharge are that...”.

R/ Done

C2/ Nothing was done!

C1/ Discussion – line 342-354: in this paragraph, you mostly discussed the literature, and not your results. That is one reason you should merge the result and discussion sections in one.

Discussion – line 447: you explained that the positive charge surface is acquired with photogenerated h+. So what about the photogenerated e-? Do you mean h+ migrate at the surface and e- toward the bulk? If yes, you should prove it. By the way, I think it is better to discuss the charge surface using pH and isoelectric point.

R/ This paragraph was rewritten to focus on pH and its relationship to the isoelectric point, which means that TiO2 at acidic pH can have a positive surface charge and plasma pre-treated plastic films have a negative surface charge.

C2/ You did not answer my comments.

C1/ Discussion – line 462: you do not discuss your results. You mentioned HO* and O2*- radicals, so you have to prove their formation by your own experimental results.

R/ It was not measured, but it was included as a possible mechanism and not as if we had tested it.

C2/ Please, you have to prove it. There is several possibilities to analyze the presence of such radicals.

C1/ Discussion – lines 476-479: you used TiO2 as photocatalyst under UVC light. So you have mainly photolysis than photocatalysis. You should do all your experiments with UVC treatment only without TiO2 since you should determine the contribution of intrinsic photocatalysis and photolysis. If you want to make TiO2 photocatalysis, you have to use UVA or UVB. So please, do it.

R/ Done

C2/ If done, where are the results?

C1/ Discussion – lines 499-509: the discussion is interesting but please, explain the plausible mechanisms.

Materials and Methods: this section is too long. It should be shortened and/or placed in supplementary information.

C2/ What about these 2 last comments?

Reviewer 2 Report

Lines 198-201. TiO2 EDS semi-quantitative chemical composition analysis revealed TiO2 was com posed of 3.19 % Titanium (atomic percentage), 96.01 % atomic percentage of oxygen, and other elements at variable proportions (Si: 0.55 % and K: 0.25 %), which make part of the substrate and surface adsorbed gases (Figure 5C). Ti + Si/O ratio was 0.04, as determined from obtained atomic data, which was below the stoichiometric value (0.5).   TiO2    contains 60 Wt. % of Ti. TiO2 cannot contain 3% of Ti!!!!! The Authors must delete this “result”. Readers are not obliged to think what does this mistake mean.

Figure 5: TiO2 (Aldrich) SEM and XRD analysis. (A) TiO2 (Aldrich) SEM images at 5,000 X TiO2 (Aldrich) (B) SEM at 212 10,000X (C) EDS analysis. (D) XRD results. (C) and (D) letters are not shown in the Figure; SEM and  EDS analysis are confused.

Author Response

C1: Lines 198-201. TiO2 EDS semi-quantitative chemical composition analysis revealed TiO2 was com posed of 3.19 % Titanium (atomic percentage), 96.01 % atomic percentage of oxygen, and other elements at variable proportions (Si: 0.55 % and K: 0.25 %), which make part of the substrate and surface adsorbed gases (Figure 5C). Ti + Si/O ratio was 0.04, as determined from obtained atomic data, which was below the stoichiometric value (0.5). TiO2 contains 60 Wt. % of Ti. TiO2 cannot contain 3% of Ti!!!!! The Authors must delete this “result”. Readers are not obliged to think what this mistake means.

 R1: Done

C1: Figure 5: TiO2 (Aldrich) SEM and XRD analysis. (A) TiO2 (Aldrich) SEM images at 5,000 X TiO2 (Aldrich) (B) SEM at 212 10,000X (C) EDS analysis. (D) XRD results. (C) and (D) letters are not shown in the Figure; SEM and EDS analysis are confused.

R1: Done

Round 3

Reviewer 1 Report

Dear Authors,

Thank you for your effort.